# Fast Lifted MAP Inference via Partitioning

**Somdeb Sarkhel**
The University of Texas at Dallas

**Parag Singla**
I.I.T. Delhi

**Vibhav Gogate**
The University of Texas at Dallas

## Abstract

Recently, there has been growing interest in lifting MAP inference algorithms for Markov logic networks (MLNs). A key advantage of these lifted algorithms is that they have much smaller computational complexity than propositional algorithms when symmetries are present in the MLN and these symmetries can be detected using lifted inference rules. Unfortunately, lifted inference rules are sound but not complete and can often miss many symmetries. This is problematic because when symmetries cannot be exploited, lifted inference algorithms ground the MLN, and search for solutions in the much larger propositional space. In this paper, we present a novel approach, which cleverly introduces new symmetries at the time of grounding. Our main idea is to partition the ground atoms and force the inference algorithm to treat all atoms in each part as indistinguishable. We show that by systematically and carefully refining (and growing) the partitions, we can build advanced any-time and any-space MAP inference algorithms. Our experiments on several real-world datasets clearly show that our new algorithm is superior to previous approaches and often finds useful symmetries in the search space that existing lifted inference rules are unable to detect.

Markov logic networks (MLNs) [5] allow application designers to compactly represent and reason about relational and probabilistic knowledge in a large number of application domains including computer vision and natural language understanding using a few weighted first-order logic formulas. These formulas act as templates for generating large Markov networks – the undirected probabilistic graphical model. A key reasoning task over MLNs is maximum a posteriori (MAP) inference, which is defined as the task of finding an assignment of values to all random variables in the Markov network that has the maximum probability. This task can be solved using propositional (graphical model) inference techniques. Unfortunately, these techniques are often impractical because the Markov networks can be quite large, having millions of variables and features.

Recently, there has been growing interest in developing lifted inference algorithms [4, 6, 17, 22] for solving the MAP inference task [1, 2, 3, 7, 13, 14, 16, 18, 19]. These algorithms work, as much as possible, on the much smaller first-order specification, grounding or propositionalizing only as necessary and can yield significant complexity reductions in practice. At a high level, lifted algorithms can be understood as algorithms that identify symmetries in the first-order specification using lifted inference rules [9, 13, 19], and then use these symmetries to simultaneously infer over multiple symmetric objects. Unfortunately, in a vast majority of cases, the inference rules are unable to identify several useful symmetries (the rules are sound but not complete), either because the symmetries are approximate or because the symmetries are domain-specific and do not belong to a known type. In such cases, lifted inference algorithms partially ground some atoms in the MLN and search for a solution in this much larger partially propositionalized space.

In this paper, we propose the following straight-forward yet principled approach for solving this *partial grounding* problem [21, 23]: partition the ground atoms into groups and force the inference algorithm to treat all atoms in each group as indistinguishable (symmetric). For example, consider a first-order atom $R(x)$ and assume that $x$ can be instantiated to the following set of constants: $\{1, 2, 3, 4, 5\}$. If the atom possesses the so-called *non-shared* or *single-occurrence* symmetry [13, 19], then the lifted inference algorithm will search over only two assignments: all five groundings of $R(x)$ are either all true or all false, in order to find the MAP solution. When no identifiable symmetries exist, the lifted algorithm will inefficiently search over all possible 32 truth assignments to the 5

ground atoms and will be equivalent in terms of (worst-case) complexity to a propositional algorithm. In our approach, we would partition the domain, say as $\{\{1,3\},\{2,4,5\}\}$, and search over only the following 4 assignments: all groundings in each part can be either all true or all false. Thus, if we are lucky and the MAP solution is one of the 4 assignments, our approach will yield significant reductions in complexity even though no identifiable symmetries exist in the problem.

Our approach is quite general and includes the fully lifted and fully propositional approaches as special cases. For instance, setting the partition size $k$ to 1 and $n$ respectively where $n$ is the number of constants will yield exactly the same solution as the one output by the fully lifted and fully propositional approach. Setting $k$ to values other than 1 and $n$ yields a family of inference schemes that systematically explores the regime between these two extremes. Moreover, by controlling the size $k$ of each partition we can control the size of the ground theory, and thus the space and time complexity of our algorithm.

We prove properties and improve upon our basic idea in several ways. First, we prove that our proposed approach yields a consistent assignment that is a lower-bound on the MAP value. Second, we show how to improve the lower bound and thus the quality of the MAP solution by systematically refining the partitions. Third, we show how to further improve the complexity of our refinement procedure by exploiting the *exchangeability* property of successive refinements. Specifically, we show that the exchangeable refinements can be arranged on a lattice, which can then be searched via a heuristic search procedure to yield an efficient any-time, any-space algorithm for MAP inference. Finally, we demonstrate experimentally that our method is highly scalable and yields close to optimal solutions in a fraction of the time as compared to existing approaches. In particular, our results show that for even small values of $k$ ($k$ bounds the partition size), our algorithm yields close to optimal MAP solutions, clearly demonstrating the power of our approach.

# 1 Notation And Background

**Partition of a Set.** A collection of sets $\mathcal{C}$ is a partition of a set $X$ if and only if each set in $\mathcal{C}$ is nonempty, pairwise disjoint and the union of all sets equals $X$. The sets in $\mathcal{C}$ are called the *cells* or *parts* of the partition. If two elements, $a, b$, of the set appear in a same cell of a partition $\rho$ we denote them by the operator '$\sim_\rho$', i.e., $a \sim_\rho b$. A partition $\alpha$ of a set $X$ is a *refinement* of a partition $\rho$ of $X$ if every element of $\alpha$ is a subset of some element of $\rho$. Informally, this means that $\alpha$ is a further fragmentation of $\rho$. We say that $\alpha$ is finer than $\rho$ (or $\rho$ is coarser than $\alpha$) and denote it as $\alpha \prec \rho$. We will also use the notation $\alpha \preceq \rho$ to denote that either $\alpha$ is finer than $\rho$, or $\alpha$ is the same as $\rho$. For example, let $\rho = \{\{1,2\},\{3\}\}$ be a partition of the set $X = \{1,2,3\}$ containing two cells $\{1,2\}$ and $\{3\}$ and let $\alpha = \{\{1\},\{2\},\{3\}\}$ be another partition of $X$, then $\alpha$ is a refinement $\rho$, namely, $\alpha \prec \rho$.

**First-order logic.** We will use a strict subset of first-order logic that has no function symbols, equality constraints or existential quantifiers. Our subset consists of (1) constants, denoted by upper case letters (e.g., $X$, $Y$, etc.), which model objects in the domain; (2) logical variables, denoted by lower case letters (e.g., $x$, $y$, etc.) which can be substituted with objects, (3) logical operators such as $\vee$ (disjunction), $\wedge$ (conjunction), $\Leftrightarrow$ (implication) and $\Rightarrow$ (equivalence), (4) universal ($\forall$) and existential ($\exists$) quantifiers and (5) predicates which model properties and relationships between objects. A predicate consists of a predicate symbol, denoted by typewriter fonts (e.g., `Friends`, `R`, etc.), followed by a parenthesized list of arguments. A *term* is a logical variable or a constant. A *literal* is a predicate or its negation. A formula in first order logic is an atom (a predicate), or any complex sentence that can be constructed from atoms using logical operators and quantifiers. For example, $\forall x$ `Smokes`$(x) \Rightarrow$ `Asthma`$(x)$ is a formula. A clause is a disjunction of literals. Throughout, we will assume that all formulas are clauses and their variables are *standardized apart*.

A *ground atom* is an atom containing only constants. A *ground formula* is a formula obtained by substituting all of its variables with a constant, namely a formula containing only ground atoms. For example, the groundings of $\neg$ `Smokes`$(x) \vee$ `Asthma`$(x)$ where $\Delta x = \{Ana, Bob\}$, are the two propositional formulas: $\neg$ `Smokes`$(Ana) \vee$ `Asthma`$(Ana)$ and $\neg$ `Smokes`$(Bob) \vee$ `Asthma`$(Bob)$.

**Markov logic.** A Markov logic network (MLN) is a set of weighted clauses in first-order logic. We will assume that all logical variables in all formulas are universally quantified (and therefore we will drop the quantifiers from all formulas), are typed and can be instantiated to a finite set of constants (for a variable $x$, this set will be denoted by $\Delta x$) and there is a one-to-one mapping between the constants and objects in the domain (Herbrand interpretations). Note that the class of MLNs we are assuming is not restrictive at all because almost all MLNs used in application domains such as

natural language processing and the Web fall in this class. Given a finite set of constants, the MLN represents a (ground) Markov network that has one random variable for each ground atom in its Herbrand base and a weighted feature for each ground clause in the Herbrand base. The weight of each feature is the weight of the corresponding first-order clause. Given a world $\omega$, which is a truth assignment to all the ground atoms, the Markov network represents the following probability distribution $P(\omega) = Z^{-1} \exp(\sum_i w_i N(f_i, \omega))$ where $(f_i, w_i)$ is a weighted first-order formula, $N(f_i, \omega)$ is the number of true groundings of $f_i$ in $\omega$ and $Z$ is the *partition function*.

For simplicity, we will assume that the MLN is in normal form, which is defined as an MLN that satisfies the following two properties: (i) there are no constants in any formula; and (ii) if two distinct atoms of predicate R have variables $x$ and $y$ as the same argument of R, then $\Delta x = \Delta y$. Because of the second condition, in normal MLNs, we can associate domains with each argument of a predicate. Let $i_R$ denote the $i$-th argument of predicate R and let $D(i_R)$ denote the number of elements in the domain of $i_R$. We will also assume that all domains are of the form $\{1, ..., D(i_R)\}$. Since domain size is finite, any domain can be converted to this form.

A common optimization inference task over MLNs is finding the most probable state of the world $\omega$, that is finding a complete assignment to all ground atoms which maximizes the probability. Formally,

$$\arg\max_\omega P_\mathcal{M}(\omega) = \arg\max_\omega \frac{1}{Z(\mathcal{M})} \exp\left(\sum_i w_i N(f_i, \omega)\right) = \arg\max_\omega \sum_i w_i N(f_i, \omega) \qquad (1)$$

From Eq. (1), we can see that the MAP problem reduces to finding a truth assignment that maximizes the sum of weights of satisfied clauses. Therefore, any weighted satisfiability solver such as MaxWalkSAT [20] can used to solve it. However, MaxWalkSAT is a propositional solver and is unable to exploit symmetries in the first-order representation, and as a result can be quite inefficient.

Alternatively, the MAP problem can be solved in a lifted manner by leveraging various lifted inference rules such as the *decomposer*, *the binomial rule* [6, 9, 22] and the recently proposed *single occurrence rule* [13, 19]. A schematic of such a procedure is given in Algorithm 1. Before presenting the algorithm, we will describe some required definitions. Let $i_R$ denote the $i$-th argument of predicate R. Given an MLN, two arguments $i_R$ and $j_S$ of its predicates R and S respectively are called *unifiable* if they share a logical variable in an MLN formula. Being symmetric and transitive, the unifiable relation splits the arguments of all the predicates into a set of *domain equivalence classes*.

**Example 1.** *Consider a normal MLN M having two weighted formulas* $(R(x) \vee S(x,y), w_1)$ *and* $(R(z) \vee T(z), w_2)$. *Here, we have two sets of domain equivalence classes* $\{1_R, 1_S, 1_T\}$ *and* $\{2_S\}$.

Algorithm 1 has five recursive steps and returns the optimal MAP value. The first two lines are the base case and the simplification step, in which the MLN is simplified by deleting redundant formulas, rewriting predicates by removing constants (so that lifted conditioning can be applied) and assigning values to ground atoms whose values can be inferred using assignments made so far. The second step is the propositional decomposition step in which the algorithm recurses over disjoint MLNs (if any) and returns their sum. In the lifted decomposition step, the algorithm finds a domain equivalence class $U$ such that in the MAP solution all ground atoms of the predicates that have elements of $U$ as arguments are either all true or all false. To find such a class, rules given in [9, 13, 19] can be used. In the algorithm, $M|U$ denotes the MLN obtained by setting the domain of all elements

| **Algorithm 1 LMAP**(MLN $M$) |
| --- |
| *// base case* |
| **if** $M$ is empty **return** 0 |
| Simplify($M$) |
| *// Propositional decomposition* |
| **if** $M$ has *disjoint* MLNs $M_1, \ldots, M_k$ **then** |
| $\quad$ **return** $\sum_{i=1}^{k}$ LMAP($M_i$) |
| *// Lifted decomposition* |
| **if** $M$ has a liftable domain equivalence class $U$ **then** |
| $\quad$ **return** LMAP($M|U$) |
| *// Lifted conditioning* |
| **if** $M$ has a *singleton atom* A **then** |
| $\quad$ **return** $\max_{i=0}^{D(1_A)}$ LMAP($M|(A, i)$) $+ w(A, i)$ |
| *// Partial grounding* |
| Heuristically select a domain equivalence class $U$ |
| and ground it yielding a new MLN $M'$ |
| **return** LMAP($M'$) |

of $U$ to 1 and updating the formula weights accordingly. In the lifted conditioning step, if there is an atom having just one argument (singleton atom), then the algorithm partitions the possible truth assignments to groundings of A such that, in each part all truth assignments have the same number of true atoms. In the algorithm, $M|(A, i)$ denotes the MLN obtained by setting $i$ groundings of A to true and the remaining to false. $w(A, i)$ is the total weight of ground formulas satisfied by the

assignment. The final step in LMAP is the *partial grounding* step and is executed only when the algorithm is unable to apply lifted inference rules. In this step, the algorithm heuristically selects a domain equivalence class $U$ and grounds it completely. For example,

**Example 2.** *Consider an MLN with two formulas: $\mathtt{R}(x,y) \vee \mathtt{S}(y,z), w_1$ and $\mathtt{S}(a,b) \vee \mathtt{T}(a,c), w_2$. Let $D(2_{\mathtt{R}}) = 2$. After grounding the equivalence class $\{2_{\mathtt{R}}, 1_{\mathtt{S}}, 1_{\mathtt{T}}\}$, we get an MLN having four formulas: $(\mathtt{R}(x_1, 1) \vee \mathtt{S}(1, z_1), w_1)$, $(\mathtt{R}(x_2, 2) \vee \mathtt{S}(1, z_2), w_1)$, $(\mathtt{S}(1, b_1) \vee \mathtt{T}(1, c_1), w_2)$ and $(\mathtt{S}(2, b_2) \vee \mathtt{T}(2, c_2), w_2)$.*[1]

## 2 Scaling up the Partial Grounding Step using Set Partitioning

Partial grounding often yields a much bigger MLN than the original MLN and is the chief reason for the inefficiency and poor scalability of Algorithm LMAP. To address this problem, we propose a novel approach to speed up inference by adding additional constraints to the existing lifted MAP formulation. Our idea is as follows: reduce the number of ground atoms by partitioning them and treating all atoms in each part as indistinguishable. Thus, instead of introducing $O(tn)$ new ground atoms where $t$ is the cardinality

---

**Algorithm 2 Constrained-Ground**
(MLN $M$, Size $k$ and domain equivalence class $U$)

$M' = M$
Create a partition $\pi$ of size $k$ of $\Delta i_{\mathtt{R}}$ where $i_{\mathtt{R}} \in U$
**foreach** predicate R such that $\exists\, i_{\mathtt{R}} \in U$ **do**
    **foreach** cell $\pi_j$ of $\pi$ **do**
        Add all possible hard formulas of the form
        $\mathtt{R}(x_1, \ldots, x_r) \Leftrightarrow \mathtt{R}(y_1, \ldots, y_r)$
        such that $x_i = y_i$ if $i_{\mathtt{R}} \notin U$ and
        $x_i = X_a, y_i = X_b$ if $i_{\mathtt{R}} \in U$ where $X_a, X_b \in \pi_j$.
**return** $M'$

---

of the domain equivalence class and $n$ is the number of constants, our approach will only introduce $O(tk)$ ground atoms where $k << n$.

Our new, approximate partial grounding method (which will replace the partial grounding step in Algorithm 1) is formally described in Algorithm 2. The algorithm takes as input an MLN $M$, an integer $k > 0$ and a domain equivalence class $U$ as input and outputs a new MLN $M'$. The algorithm first partitions the domain of the class $U$ into $k$ cells, yielding a partition $\pi$. Then, for each cell $\pi_j$ of $\pi$ and each predicate R such that one or more of its arguments is in $U$, the algorithm adds all possible constraints of the form $\mathtt{R}(x_1, \ldots, x_r) \Leftrightarrow \mathtt{R}(y_1, \ldots, y_r)$ such that for each $i$: (1) we add the equality constraint between the logical variables $x_i$ and $y_i$ if the $i$-th argument of the predicate is not in $U$ and (1) set $x_i = X_a$ and $y_i = X_b$ if $i$-th argument of R is in $U$ where $X_a, X_b \in \pi_j$. Since adding constraints restricts feasible solutions to the optimization problem, it is easy to show that:

**Proposition 1.** *Let $M' = \textbf{Constrain-Ground}(M, k)$, where $M$ is an MLN and $k > 0$ is an integer, be the MLN used in the partial grounding step of Algorithm 1 (instead of the partial grounding step described in the algorithm). Then, the MAP value returned by the modified algorithm will be smaller than or equal to the one returned by Algorithm 1.*

The following example demonstrates how Algorithm 2 constructs a new MLN.

**Example 3.** *Consider the MLN in Example 2. Let $\{\{1, D_{2,R}\}\}$ be a 1-partition of the domain of $U$. Then, after applying Algorithm 2, the new MLN will have the following three hard formulas in addition to the formulas given in Example 2: (1) $\mathtt{R}(x_3, 1) \Leftrightarrow \mathtt{R}(x_3, 2)$, (2) $\mathtt{S}(1, x_4) \Leftrightarrow \mathtt{S}(2, x_4)$ and (3) $\mathtt{T}(1, x_5) \Leftrightarrow \mathtt{T}(2, x_5)$.*

Although, adding constraints reduces the search space of the MAP problem, Algorithm 2 still needs to ground the MLN. This can be time consuming. Alternatively, we can group indistinguishable atoms together without grounding the MLN using the following definition:

**Definition 1.** *Let $U$ be a domain equivalence class and let $\pi$ be its partition. Two ground atoms $\mathtt{R}(x_1, ..., x_r)$ and $\mathtt{R}(y_1, ..., y_r)$ of a predicate R such that $\exists i_{\mathtt{R}} \in U$ are **equivalent** if $x_i = y_i$ if $i_{\mathtt{R}} \notin U$ and $x_i = X_a, y_i = X_b$ if $i_{\mathtt{R}} \in U$ where $X_a, X_b \in \pi_j$. We denote this by $\mathtt{R}(x_1, ..., x_r) \perp_{\pi} \mathtt{R}(y_1, ..., y_r)$.*

Notice that the relation $\perp_{\pi}$ is symmetric and reflexive. Thus, we can group all the ground atoms corresponding to the transitive closure of this relation, yielding a "meta ground atom" such that if the meta atom is assigned to true (false), all the ground atoms in the transitive closure will be true (false). This yields the **partition-ground** algorithm described as Algorithm 3. The algorithm starts

by creating a $k$ partition of the domain of $U$. It then updates the domain of $U$ so that it only contains $k$ values, grounds all arguments of predicates that are in the set $U$ and updates the formula weights appropriately. The formula weights should be updated because, when the domain is compressed, several ground formulas are replaced by just one ground formula. Intuitively, if $t$ (partially) ground formulas having weight $w$ are replaced by one (partially) ground formula $(f, w')$ then $w'$ should be equal to $wt$. The two for loops in Algorithm 3 accomplish this. We can show that:

**Proposition 2.** *The MAP value output by replacing the partial grounding step in Algorithm 1 with Algorithm Partition-Ground, is the same as the one output by replacing the the partial grounding step in Algorithm 1 with Algorithm Constrained-Ground.*

The key advantage using Algorithm Partition-Ground is that the lifted algorithm (LMAP) will have much smaller space complexity than the one using Algorithm Constrained-Ground. Specifically, unlike the latter, which yields $O(n|U|)$ ground atoms (assuming each predicate has only one argument in $U$) where $n$ is the number of constants in the domain of $U$, the former generates only $O(k|U|)$ ground atoms, where $k << n$.

The following example illustrates how algorithm partition-ground constructs a new MLN.

---

**Algorithm 3 Partition-Ground**
(MLN $M$, Size $k$ and domain equivalence class $U$)

---

$M' = M$

Create a partition $\pi$ of size $k$ of $\Delta i_{\mathtt{R}}$ where $i_{\mathtt{R}} \in U$

Update the domain $\Delta i_{\mathtt{R}}$ to $\{1, \ldots, k\}$ in $M'$

Ground all predicates $\mathtt{R}$ such that $i_{\mathtt{R}} \in U$

**foreach** formula $(f', w')$ in $M'$ such that $f$

contains an atom of $\mathtt{R}$ where $i_{\mathtt{R}} \in U$ **do**

    Let $f$ be the formula in $M$ from which $f'$ was derived

    **foreach** logical variable in $f$ that was substituted

    by the $j$-th value in $\Delta i_{\mathtt{R}}$ to yield $f'$ **do**

        $w' = w' \times |\pi_j|$ where $\pi_j$ is the $j$-th cell of $\pi$

**return** $M'$

---

**Example 4.** *Consider an MLN $M$, with two formulas: $(\mathtt{R}(x, y) \lor \mathtt{S}(y, z), w_1)$ and $(\mathtt{S}(a, b) \lor \mathtt{T}(a, c), w_2)$. Let $D(2_{\mathtt{R}}) = 3$ and $\pi = \{\{1, 2\}, \{3\}\} = \{\nu_1, \nu_2\}$. After grounding $2_{\mathtt{R}}$ with respect to $\pi$, we get an MLN, $M'$, having four formulas: $(\mathtt{R}_{\nu_1}(x_1) \lor \mathtt{S}_{\nu_1}(z_1), 2w_1)$, $(\mathtt{R}_{\nu_2}(x_2) \lor \mathtt{S}_{\nu_2}(z_2), w_1)$, $(\mathtt{S}_{\nu_1}(b_1) \lor \mathtt{T}_{\nu_1}(c_1), 2w_2)$ and $(\mathtt{S}_{\nu_2}(b_2) \lor \mathtt{T}_{\nu_2}(c_2), w_2)$. The total weight of grounding in $M$ is $(3w_1 D(1_{\mathtt{R}}) D(2_{\mathtt{S}}) + 3w_2 D(2_{\mathtt{T}}) D(2_{\mathtt{S}}))$ which is the same as in $M'$.*

The following example illustrates how the algorithm constructs a new MLN in presence of self-joins.

**Example 5.** *Consider an MLN, $M$, with the single formula: $\neg \mathtt{R}(x, y) \lor \mathtt{R}(y, x), w$. Let $D(1_{\mathtt{R}}) = D(2_{\mathtt{R}}) = 3$ and $\pi = \{\{1, 2\}, \{3\}\} = \{\nu_1, \nu_2\}$. After grounding $1_{\mathtt{R}}$ (and also on $D(2_{\mathtt{R}})$, as they belong to the same domain equivalence class) with respect to $\pi$, we get an MLN, $M'$, having following four formulas: $(\mathtt{R}_{\nu_1, \nu_1} \lor \mathtt{R}_{\nu_1, \nu_1}, 4w)$, $(\mathtt{R}_{\nu_1, \nu_2} \lor \mathtt{R}_{\nu_2, \nu_1}, 2w)$, $(\mathtt{R}_{\nu_2, \nu_1} \lor \mathtt{R}_{\nu_1, \nu_2}, 2w)$ and $(\mathtt{R}_{\nu_2, \nu_2} \lor \mathtt{R}_{\nu_2, \nu_2}, w)$.*

### 2.1 Generalizing the Partition Grounding Approach

Algorithm Partition-Ground allows us to group the equivalent atoms with respect to a partition and has much smaller space complexity and time complexity than the partial grounding strategy described in Algorithm 1. However, it yields a lower bound on the MAP value. In this section, we show how to improve the lower bound using refinements of the partition. The basis of our generalization is the following theorem:

**Theorem 1.** *Given two partitions $\pi$ and $\phi$ of $U$ such that $\phi \preceq \pi$, the MAP value of the partially ground MLN with respect to $\phi$ is less than or equal to the MAP value of the partially ground MLN with respect to $\pi$ .*

*Proof. Sketch:* Since the partition $\phi$ is a finer refinement of $\pi$, any candidate MAP assignment corresponding to the MLN obtained via $\phi$ already includes all the candidate assignments corresponding to the MLN obtained via $\pi$, and since the MAP value of both of these MLNs are a lower bound of the original MAP value, the theorem follows. $\square$

We can use Theorem 1 to devise a new any-time MAP algorithm which refines the partitions to get a better estimate of MAP values. Our approach is presented in Algorithm 4.

The algorithm begins by identifying all non-liftable domains, namely domains $U_i$ that will be partially grounded during the execution of Algorithm 1, and associating a 1-partition $\pi_i$ with each domain. Then, until there is timeout, it iterates through the following two steps. First, it runs the LMAP algorithm, which uses the pair $(U_i, \pi_i)$ in Algorithm partition-ground during the $i$-th partial

grounding step, yielding a MAP solution $\mu$. Second, it heuristically selects a partition $\pi_j$ and refines it. From Theorem 1, it is clear that as the number of iterations is increased, the MAP solution will either improve or remain the same. Thus, Algorithm Refine-MAP is an anytime algorithm.

Alternatively, we can also devise an any-space algorithm using the following idea. We will first determine $k$, the maximum size of a partition that we can fit in the memory. As different partitions of size $k$ will give us different MAP values, we can search through them to find the best possible MAP solution. A drawback of the any-space approach is that it explores a prohibitively large search space. In particular, the number of possible partitions of size $k$ for a set of size $n$ (denoted by $\left\{ {n \atop k} \right\}$) is given by the so called *Stirling numbers of the second*

---

**Algorithm 4 Refine-MAP**(MLN $M$)

Let $\mathcal{U} = \{U_i\}$ be the non-liftable domains
Set $\pi_i = \{\Delta j_\text{R}\}$ where $j_\text{R} \in U_i$ for all $U_i \in \mathcal{U}$
$\mu = -\infty$
**while** timeout has not occurred **do**
   $\mu = $LMAP$(M)$
   */\* LMAP uses the pair $(U_i, \pi_i)$ and Algorithm partition-ground for its i-th partial grounding step. \*/*
   Heuristically select a partition $\pi_j$ and refine it
**return** $\mu$

---

*kind* which grows exponentially with $n$. (The total number of partitions of a set is given by the *Bell number*, $B_n = \sum_{k=1}^{n} \left\{ {n \atop k} \right\}$). Clearly, searching over all the possible partitions of size $k$ is not practical. Luckily, we can exploit symmetries in the MLN representation to substantially reduce the number of partitions we have to consider, since many of them will give us the same MAP value. Formally,

**Theorem 2.** *Given two $k$-partitions $\pi = \{\pi_1, \ldots, \pi_k\}$ and $\phi = \{\phi_1, \ldots, \phi_k\}$ of $U$ such that $|\pi_i| = |\phi_i|$ for all $i$, the MAP value of the partially ground MLN with respect to $\pi$ is equal to the MAP value of the partially ground MLN with respect to $\phi$.*

*Proof. Sketch:* A formula $f$, when ground on an argument $i_\text{R}$ with respect to a partition $\pi$ creates $|\pi|$ copies of the formula. Since $|\phi| = |\pi| = k$ grounding on $i_\text{R}$ with respect to $\phi$ also creates the same number of formulas which are identical upto a renaming of constants. Furthermore, since $|\pi_i| = |\phi_i|$ (each of their parts have identical cardinality) and as weight of a ground formula is determined by the cell sizes (see Algorithm Partition-Ground) the ground formulas obtained using $\phi$ and $\pi$ will have same weights as well. As a result, MLNs obtained by grounding on any argument $i_\text{R}$ with respect to $\phi$ and $\pi$ are indistinguishable (subject to renaming of variables and constants) and the proof follows. $\square$

From Theorem 2, it follows that the number of elements in cells and the number of cells of a partition is sufficient to define a partially ground MLN with respect to that partition. Consecutive refinements of such partitions will thus yield a lattice, which we will refer to as *Exchangeable Partition Lattice*. The term 'exchangeable' refers to the fact that two partitions containing same number of elements of cells and same number of cells are exchangeable with each other (in terms of MAP so-

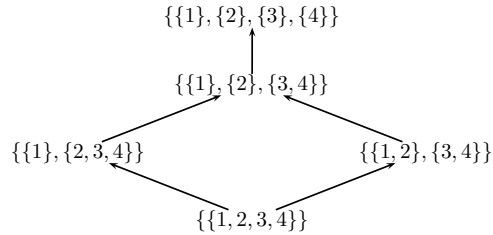

Figure 1: *Exchangeable Partition Lattice* corresponding to the domain $\{1, 2, 3, 4\}$.

lution quality). Figure 1 shows the Exchangeable Partition Lattice corresponding to the domain $\{1, 2, 3, 4\}$. If we do not use exchangeability, the number of partitions in the lattice would have been $B_4 = \left\{ {4 \atop 1} \right\} + \left\{ {4 \atop 2} \right\} + \left\{ {4 \atop 3} \right\} + \left\{ {4 \atop 4} \right\} = 1 + 7 + 6 + 1 = 15$. On the other hand, the lattice has 5 elements.

Different traversal strategies of this exchangeable partition lattice will give rise to different lifted MAP algorithms. For example, a greedy depth-first traversal of the lattice yields Algorithm 4. We can also explore the lattice using systematic depth-limited search and return the maximum solution found for a particular depth limit $d$. This yields an improved version of our any-space approach described earlier. We can even combine the two strategies by traversing the lattice in some heuristic order. For our experiments, we use greedy depth-limited search, because full depth-limited search was very expensive. Note that although our algorithm assumes normal MLNs, which are pre-shattered, we can easily extend it to use shattering as needed [10]. Moreover by clustering evidence atoms together [21, 23] we can further reduce the size of the shattered theory [4].

# 3 Experiments

We implemented our algorithm on top of the lifted MAP algorithm of Sarkhel et al. [18], which reduces lifted MAP inference to an integer polynomial program (IPP). We will call our algorithm P-IPP (which stands for partition-based IPP). We performed two sets of experiments. The first set measures the impact of increasing the partition size $k$ on the quality of the MAP solution output by our algorithm. The second set compares the performance and scalability of our algorithm with several algorithms from literature. All of our experiments were run on a third generation i7 quad-core machine having 8GB RAM.

We used following five MLNs in our experimental study: (1) An MLN which we call **Equivalence** that consists of following three formulas: Equals($x$,$x$), Equals($x$,$y$) $\rightarrow$ Equals($y$,$x$), and Equals($x$,$y$) $\wedge$ Equals($y$,$z$) $\rightarrow$ Equals($x$,$z$); (2) The **Student** MLN from [18, 19], consisting of four formulas and three predicates; (3) The **Relationship** MLN from [18], consisting of four formulas and three predicates; (4) **WebKB** MLN [11] from the Alchemy web page, consisting of three predicates and seven formulas; and (5) **Citation Information-Extraction** (IE) MLN from the Alchemy web page [11], consisting of five predicates and fourteen formulas .

We compared the solution quality and scalability of our approach with the following algorithms and systems: Alchemy (ALY) [11], Tuffy (TUFFY) [15], ground inference based on integer linear programming (ILP) and the IPP algorithm of Sarkhel et al. [18]. Alchemy and Tuffy are two state-of-the-art open source software packages for learning and inference in MLNs. Both of them ground the MLN and then use an approximate solver, MaxWalkSAT [20] to compute the MAP solution. Unlike Alchemy, Tuffy uses clever Database tricks to speed up computation and in principle can be much more scalable than Alchemy. ILP is obtained by converting the MAP problem over the ground Markov network to an Integer Linear Program. We ran each algorithm on the aforementioned MLNs for varying time-bounds and recorded the solution quality, which is measured using the total weight of the false clauses in the (approximate) MAP solution, also referred to as the cost. Smaller the cost, better the MAP solution. For a fair comparison, we used a parallelized Integer Linear Programming solver called Gurobi [8] to solve the integer linear programming problems generated by our algorithm as well as by other competing algorithms.

Figure 2 shows our experimental results. Note that if the curve for an algorithm is not present in a plot, then it means that the corresponding algorithm ran out of either memory or time on the MLN and did not output any solution. We observe that Tuffy and Alchemy are the worst performing systems both in terms of solution quality and scalability. ILP scales slightly better than Tuffy and Alchemy. However, it is unable to handle MLNs having more than 30K clauses. We can see that our new algorithm P-IPP, run as an anytime scheme, by refining partitions, not only finds higher quality MAP solutions but also scales better in terms of time complexity than IPP. In particular, IPP could not scale to the equivalence MLN having roughly 1 million ground clauses and the relation MLN having roughly 125.8M ground clauses. The reason is that these MLNs have self-joins (same predicate appearing multiple times in a formula), which IPP is unable to lift. On the other hand, our new approach is able to find useful approximate symmetries in these hard MLNs.

To measure the impact of varying the partition size on the MAP solution quality, we conducted the following experiment. We first ran the IPP algorithm until completion to compute the optimum MAP value. Then, we ran our algorithm multiple times, until completion as well, and recorded the solution quality achieved in each run for different partition sizes. Figure 3 plots average cost across various runs as a function of $k$ (the error bars show the standard deviation). For brevity, we only show results for the IE and Equivalence MLNs. The optimum solutions for the three MLNs were found in (a) 20 minutes, (b) 6 hours and (c) 8 hours respectively. On the other hand, our new approach P-IPP yields close to optimal solutions in a fraction of the time, and for relatively small values of $k$ ($\approx 5 - 10$).

# 4 Summary and Future Work

Lifted inference techniques have gained popularity in recent years, and have quickly become the approach of choice to scale up inference in MLNs. A pressing issue with existing lifted inference technology is that most algorithms only exploit exact, identifiable symmetries and resort to grounding or propositional inference when such symmetries are not present. This is problematic because grounding can blow up the search space. In this paper, we proposed a principled, approximate approach to solve this grounding problem. The main idea in our approach is to partition the ground atoms into a small number of groups and then treat all ground atoms in a group as indistinguishable

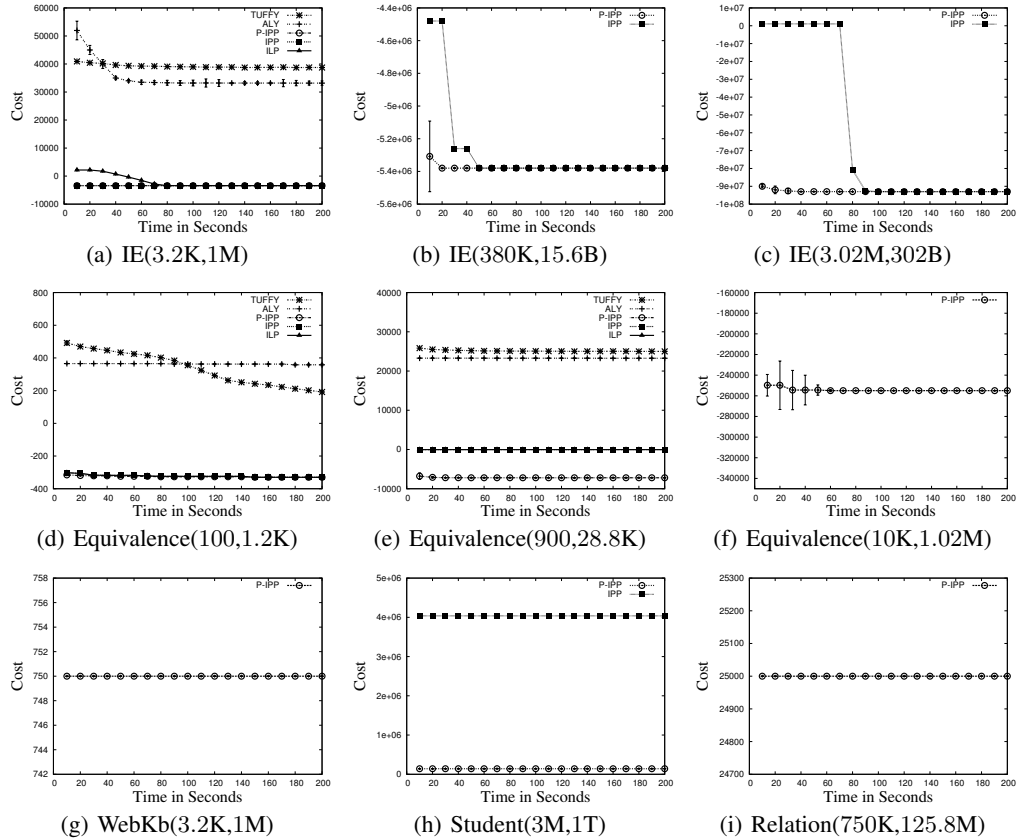

Figure 2: Cost vs Time: Cost of unsatisfied clauses(smaller is better) vs time for different domain sizes. Notation used to label each figure: MLN(numvariables, numclauses). Note: the quantities reported are for ground Markov network associated with the MLN. Standard deviation is plotted as error bars.

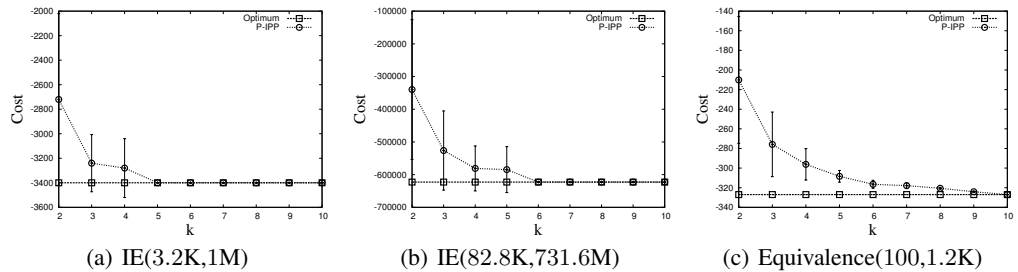

Figure 3: Cost vs Partition Size: Notation used to label each figure: MLN(numvariables, numclauses).

(from each other). This simple idea introduces new, approximate symmetries which can help speed-up the inference process. Although our proposed approach is inherently approximate, we proved that it has nice theoretical properties in that it is guaranteed to yield a consistent assignment that is a lower-bound on the MAP value. We further described an any-time algorithm which can improve this lower bound through systematic refinement of the partitions. Finally, based on the exchangeability property of the refined partitions, we demonstrated a method for organizing the partitions in a lattice structure which can be traversed heuristically to yield efficient any-time as well as any-space lifted MAP inference algorithms. Our experiments on a wide variety of benchmark MLNs clearly demonstrate the power of our new approach. Future work includes connecting this work to the work on Sherali-Adams hierarchy [2]; deriving a variational principle for our method [14]; and developing novel branch and bound [12] as well as weight learning algorithms based on our partitioning approach.

**Acknowledgments:** This work was supported in part by the DARPA Probabilistic Programming for Advanced Machine Learning Program under AFRL prime contract number FA8750-14-C-0005.

## Footnotes

[1]The constants can be removed by renaming the predicates yielding a normal MLN. For example, we can rename $\mathtt{R}(x_1, 1)$ as $\mathtt{R}_1(x_1)$. This renaming occurs in the simplification step.

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
