[Reviews · NeurIPS 2015]

Submitted by Assigned_Reviewer_1

The paper considers lifted MAP inference for relational models.

More precisely, it proposes to make use of partitions and partition lattices for symmetry-aware MAP inference.

Overall, I very much like the direction of the paper. And the experimental results shows that approach works well; some of the

results are actually quite impressive.

However, there are also some downsides:

(1) The ideas exploited are floating in the literature and this should be acknowledged (2) The baselines in the experimental evaluation are rather weak; at least the rocket system should be used as another baseline

Let me clarify this in more details.

The paper is too short on related work. The discussion of the related work has to

be extended. At the current stage, it is not clear where the

presented work goes beyond what is known, in particular when it comes to algebraic lifted MAP approaches. The long list of references on the first page is not very informative there. More importantly,

it is not even clear what "in a vast majority of cases, the inference

rules are unable to identify several useful symmetries" actually means running the risk of providing false statements about some of the other approaches. That is, the connections are not formally defined so far in the literature and without this, the claim is not proven at all.

Moreover, many of the positive claims made about the presented work

also holds for the algebraic approaches as presented in [2, 13, 15]

and also

Martin Mladenov, Amir Globerson, Kristian Kersting: Lifted Message Passing as Reparametrization of Graphical Models

UAI 2014

Martin Mladenov, Kristian Kersting:

Equitable Partitions of Concave Free Energies.

UAI 2015

All of them show that we can solve the relaxed MAP LP

efficiently using (fractional) automorphisms. Plugging them into

a brunch-and-cut/bound ILP solver would result in a very similar if not identical approach, modulo the efficient recompilation of the lifted network and partial grounding. They could be

combined with SA hierarchies as e.g. done in

Udi Apsel, Kristian Kersting, Martin Mladenov: Lifting Relational MAP-LPs Using Cluster Signatures.

AAAI 2014

Moreover, the domain approach presented in the present paper appears to be similar if not identical to the domain graph approach presented in some of the above mentioned related approaches. The authors should clarify the differences.

Indeed, one of the big differences is the refinement of the lifted during inference. However, much of the lattice approach is already

presented in the above mentioned work, indeed from an algebraic

perspective and not from a logical perspective, but this is a detail.

Moreover, the idea of relifting during inference has been presented already in [7].

That is, the paper misses a discussion of related work. This should carefully acknowledge the origins of ideas and in turn showecase the originality of the presented idea. Without, it reads more like putting well-known ideas together.

Nevertheless, putting the ideas together into a single system is really nice and scales inference very well.
Summary: + interesting combination of different ideas mentioned in the literature + shows great empirical results

- very weak on related work; actually so weak that it is hard if not impossible to judge the originality of the presented work. - low originality

Submitted by Assigned_Reviewer_2

This paper presents an approximate lifted MAP inference algorithm for Markov logic networks (MLNs). Main contribution of this paper is to reduce the number of ground atoms by heuristic partitioning and assuming each part as indistinguishable. As authors mentioned, the exchangeability assumption (approximation) can be handled by a partition refinement (as if, there are only singleton sets and not exchangeability assumption) as authors explain in Theorem 1.

I believe that this is a good step forward for practical lifted MAP inference. Also, authors explains the main idea of this paper well in compared to existing literature.

Here are my suggestion and questions for authors for possible rebuttals/revisions.

1. The main theoretical contribution seems not unexpected in general. I mean that the difficulty of lifted MAP inference comes the fact that many existing lifted inference rules are sound but not complete. If authors want to overcome this difficulty by approximate partition (exchangeability assumption), an interesting questions would be (1) how much the approximate MAP value deviate from the true MAP one, and (2) when we can (cannot) ignore those errors.

2. Personally, I feel that topics of this paper is too general when describing algorithms. As an example, Algorithm 1 describes previous work. However, for me, it is not clear which heuristic selection algorithms (in partial grounding) that authors are referring to. Since, Proposition 1 and 2 are written in compared to the Algorithm 1. It should be clarified. In this regard, it is not also clear how Algorithms 2 and 3 create a partition \pi of size k. Since there would be numerous cases to partition a domain, it would be helpful if authors can provide more detailed descriptions of the partitioning process.

Summary: This paper presents an approximate lifted MAP inference algorithm for Markov logic networks (MLNs). Although the presented algorithms show the theoretical potential of the new proposed algorith, this paper may need more clarifications before published.

Submitted by Assigned_Reviewer_3

This paper proposes a smart idea for improving MAP computation in MLN: basically, one splits the set of MLNs until finding a good compromise between approximation and computation. The price is always having a lower bound, with no guarantees on quality.

The idea is strange at first, but results look good. Comparison with related work is weak.

Comments:

paragraph 049: this is confusing -> do you partition the atoms or the domain? It is not quite the same.

line 136: add a break here, is this version yours or are you just following literature?

line 175: can you quickly explain why is this the case and why it cannot be avoided? Your point seems to be that a lifted algorithm can be(arbitrarily) worse than a grounded one

211: Def 1 seems to be very similar to lifted BP

What do you mean by "Cost of unsatisfied clauses", please explain in the text.

Related Work: please explain why your solution is better in some more detail.
Summary: This paper proposes a smart idea for improving MAP computation in MLN: basically, one splits the set of MLNs until finding a good compromise between approximation and computation. The price is always having a lower bound, with no guarantees on quality.

The idea is strange at first, but results look good. Comparison with related work is weak.

Author Feedback
Author rebuttal: Thanks to all the reviewers for their helpful comments.

Reviewer_1

Although we have used [18] for our experiments, it should be noted that our approach is applicable to any exact approach that performs partial grounding (even the algebraic approaches mentioned in your review).

The purpose of the statement "in vast majority ..." is to emphasize that lifted inference rules are sound but not complete.

On the surface our approach does look similar to the SA hierarchy, but note that we are starting with an exact formulation (and not a relaxed LP formulation). Therefore, any solution generated from our approach is a feasible solution. In fact, our approach is polar opposite of the SA approach and thus complementary. The initial ILP is difficult to solve and the constraints are added in Algorithm 2 to reduce the search space and make inference easy. As a result, the solution obtained after adding the constraints will be inferior to the one obtained from the initial ILP formulation (as opposed to better in case of SA). Moreover, as we go down a single path along the refinement lattice, the constraints are not added, but removed to give us a tighter lower bound and a better feasible solution (as opposed to tighter upper bound in SA). Algorithm 2 is presented only to simplify our presentation. As we describe, we use Algorithms 3 and 4 in our experiments. The main inspiration behind Algorithm 2 is constraint satisfaction and branch and bound literature where constraints are often used to reduce search spaces.

Notice that we are using more powerful lifted inference rules than [2,3,15] and Mladenov papers (for example, the lifted conditioning rule and MAP-specific rules in [18] are not fully utilized there). Our ILP is much smaller than the one they relax and thus may lead to better SA hierarchy but that is a topic of future investigation.

We believe our main contribution indeed is the refinement architecture and realization that many of the possible partitions are in fact indistinguishable from each other (Theorem 2).

We will clarify the related work if the paper is accepted and also include a discussion on how it can be combined with the SA hierarchy in a separate future work section.

Reviewer_2

Regarding your two theoretical questions (1) how much the approximate MAP value deviate from the true MAP one, and (2) when we can(not) ignore errors. These questions are not that easy to answer because even analyzing the cases at level 2 and level 3 of the lattice is a hard problem. They include problems such as the parity problem and 3-partite perfect matching problem as special cases. Therefore, we have investigated the questions empirically. In our experiments, we show that we don't have to explore the full lattice to get optimal or nearly optimal solutions; just do a simple depth-bounded search over it. We can easily derive bounds by distributing max over sums but again they will be very loose. We hope this changes your mind about the utility of the results presented in the paper.

You have correctly identified that there can be numerous ways of partitioning a domain (as stated in the paragraph before theorem 2 and as illustrated next to figure 1). However, as stated in theorem 2 many of these partitions are in fact indistinguishable with each other (in terms of MAP solution quality). One obvious strategy is to determine K, the maximum size of a partition that we can fit in the memory and search through all of them (mentioned in the paragraph just before theorem 2). But for our experiments we have used a random depth bounded search over the exchangeable partition lattice (mentioned in the last paragraph of section 3). We will discuss the heuristics at end of section 2 if the paper is accepted; note however that our theorems do not depend on the heuristics used.

Reviewer_3

We will fix all the typos in the camera-ready version

We partition the domains. As a result of the partition, some ground atoms become indistinguishable. We will clarify this.

The main reason for the poor scalability of lifted algorithms is that the lifted rules are sound but not complete and in general cannot exploit several useful symmetries present in the relational model. Although lifted algorithms might not perform worse than propositional algorithms, they are still not scalable enough due to the large size of the ground theory.

MAP is a maximization problem, which in case of MLNs is computed as sum of the weights of satisfied clauses. However, in literature, optimization problems are generally formulated as minimization problem. The minimization formulation of MAP is computed by sum of the weights of unsatisfied clauses. This is what we have mentioned as the cost. In the camera-ready version we will explicitly mention this.

Related work: please refer above.

Reviewer_5
We will reword footnote 1. We will remove the word ad-hoc and the last sentence. (It does sound terrible after a second read).